# Cholinergic Modulation of Neuroinflammation: Focus on α7 Nicotinic Receptor

**DOI:** 10.3390/ijms22094912

**Published:** 2021-05-06

**Authors:** Roberta Piovesana, Michael Sebastian Salazar Intriago, Luciana Dini, Ada Maria Tata

**Affiliations:** 1Département de Neurosciences, Université de Montréal, Montréal, QC H3C 3J7, Canada; roberta.piovesana@umontreal.ca; 2Groupe de Recherche sur le Système Nerveux Central, Université de Montréal, Montréal, QC H3C 3J7, Canada; 3Department of Biology and Biotechnologies “Charles Darwin”, Sapienza, University of Rome, 00185 Rome, Italy; michaelsebastian.salazarintriago@uniroma1.it (M.S.S.I.); luciana.dini@uniroma1.it (L.D.); 4Research Centre of Neurobiology “Daniel Bovet”, Sapienza, University of Rome, 00185 Rome, Italy

**Keywords:** α7 nicotinic receptors, acetylcholine, glial cells, neuroinflammation, metabotropic signalling

## Abstract

All nervous system pathologies (e.g., neurodegenerative/demyelinating diseases and brain tumours) develop neuroinflammation, a beneficial process during pathological events, aimed at removing damaged cells, toxic agents, and/or pathogens. Unfortunately, excessive inflammation frequently occurs during nervous system disorders, becoming a detrimental event capable of enhancing neurons and myelinating glial cell impairment, rather than improving their survival and activity. Consequently, targeting the neuroinflammation could be relevant for reducing brain injury and rescuing neuronal and glial cell functions. Several studies have highlighted the role of acetylcholine and its receptors in the regulation of central and peripheral inflammation. In particular, α7 nicotinic receptor has been described as one of the main regulators of the “brain cholinergic anti-inflammatory pathway”. Its expression in astrocytes and microglial cells and the ability to modulate anti-inflammatory cytokines make this receptor a new interesting therapeutic target for neuroinflammation regulation. In this review, we summarize the distribution and physiological functions of the α7 nicotinic receptor in glial cells (astrocytes and microglia) and its role in the modulation of neuroinflammation. Moreover, we explore how its altered expression and function contribute to the development of different neurological pathologies and exacerbate neuroinflammatory processes.

## 1. Introduction

Inflammation is a critical and indispensable physiological process that coordinates immune response and drives tissue regeneration after trauma or infection, activating different signalling pathways that lead to the recruitment of inflammatory cells such as neutrophils and macrophages. Neuroinflammation in the central nervous system (CNS) includes a dynamic multistage physiological response directed by the CNS glial cells (microglia and astrocytes). The innate immune system is classically activated in neurodegenerative diseases, such as spinal cord injury, multiple sclerosis, brain injury, and tumours, and activates a complex cascade of pro-inflammatory factors that modify the CNS microenvironment in response to damage. Several studies have described how the neurotransmitter acetylcholine (ACh) modulates the production of multiple inflammatory cytokines both in the immune system and in the brain [1,2], being its receptors and enzymes not only restricted to cholinergic neurons, but also expressed by many non-neuronal cells. Evidence supporting ACh involvement in the control of inflammation was the discovery of the cholinergic control on pro-inflammatory cytokine production in macrophages [2]. Experiments performed using ACh mimetics, such as muscarine and nicotine, have demonstrated a significant inhibition of pro-inflammatory cytokine production (i.e., IL-1β) only after the nicotine treatment in human primary macrophages. Moreover, macrophages derived from α7 knockout mice do not respond to cholinergic agonists, and tumour necrosis factor (TNF) production is not counteracted by the presence of either nicotine or ACh [2], enhancing the idea that cholinergic anti-inflammatory action is mediated by α7 nicotinic receptor subtype (nAChR) [2]. Although multiple subunits were found in several immune cells, a central contribution of α7 nAChRs in the cholinergic regulation of inflammatory response was confirmed [3]. In CNS, the electrical stimulation of the vagus nerve shows a significant reduction of endotoxin-induced serum TNF levels in wild-type mice, whereas this fails in α7-deficient mice [2]. ACh also binds α7 nAChRs expressed by microglia and astrocytes with a reduction of neuroinflammation. The “cholinergic anti-inflammatory pathway” and its role in immunity and neuroinflammation have gained considerable attention due to α7 nAChR alteration correlating with several human pathologies, including sepsis, diabetes, osteoarthritis, inflammatory bowel disease, and neuropsychiatric and neurological disorders. In this review, we report the involvement of α7 nAChR in neuroinflammation and in CNS pathologies. Moreover, we emphasize the new α7 nAChR selective agonists, which may have a potential clinical use in the treatment of neuroinflammation.

## 2. α7 nAChR Subtype

nAChRs are one of two classes of cholinergic receptors able to bind to ACh. nAChRs are non-selective cation channels that conduct Na^+^, K^+^, and Ca^2+^ subsequently to the ligand binding (Figure 1). The different nAChR subtypes are widely expressed from nematodes to humans, and are distributed in many regions of the central and peripheral nervous system [4] and in the immune cells [1,5,6]. nAChRs are organized as hetero- or homo-pentamers, forming the receptor channel. The different nicotinic receptor functions depend on the possible combinations of these subunits. The loop connecting the transmembrane (TM) regions, TM2 and TM3, contributes to the mechanism of receptor activation [7,8,9]. α7 nAChR is one of the main receptor subtypes expressed in the brain, where it has a central role in brain development and also in adult brain functioning such as learning, synaptic plasticity, memory, locomotion, attention, and anxiety [3,10]. It is a homo-pentamer consisting of five identical α7 subunits (Figure 1) [11,12]. Compared to other nicotinic subtypes, α7 exhibits unique functional characteristics, including:-high permeability to calcium;-rapid activation and desensitization phase (in the order of milliseconds);-selective inhibition given by α-bungarotoxin (α-BTX) and methyllycaconitine (MLA), and a low affinity for nicotine.

Recently, it has been proposed that in addition to rapid responses of membrane depolarization induced by inward currents via ion channels, nAChRs can generate longer-lasting neuronal effects, where rapid cation influx may also activate different intracellular signalling transduction pathways through G protein coupling (Figure 1) [13,14,15]. In neurons, both ionotropic and metabotropic α7 nAChR functions are observed [16].

## 3. α7 Metabotropic Signalling Mediates the Cholinergic Anti-Inflammatory Pathway

The ionotropic activity of α7 nAChR is a specific property of neurons. The activation of ionotropic α7 nAChRs involves a wide range of Ca^2+^ sensitive targets, including enzymes such as cyclic-dependent AMP protein kinase (PKA) and Ca^2+^/calmodulin-dependent protein kinase. These kinases, sensitive to intracellular Ca^2+^ levels, can regulate various synaptic ion channels, as well as cytoskeletal and trafficking proteins, which control vesicle mobility and release [16]. Furthermore, calcium cell signalling mediated by nAChRs regulates gene expression in neurons, controlling the activation of transcription factors such as CREB, which plays an important role in memory and learning [17].

In non-neuronal cells (i.e., immune cells and glial cells), the metabotropic ways are prevalent downstream of α7 nAChR activation, and have largely been characterized. In mouse macrophages, α7 nAChR mediates anti-inflammatory response through the involvement of two intracellular signalling pathways: Jak2/STAT3 and PI3K/Akt [18,19]. These metabotropic signalling pathways result in the negative modulation of the nuclear factor (NF)-kB, responsible for pro-inflammatory cytokine expression (i.e., TNFα, IL-1β, IL-6) and the positive modulation of nuclear factor erythroid 2–related factor 2 (Nrf2) [3] (Figure 2A). It has been observed that the activation of α7 nAChR mediates the recruitment and phosphorylation of Janus kinase 2 (Jak2) [18]. Once activated, Jak2 phosphorylates and activates the signal transducer and activator of transcription 3 (STAT3), blocking the translocation of NF-kB into the cell nucleus and the subsequent NF-kB binding to DNA (Figure 2A) [3,18]. NF-kB is a transcriptional factor capable of coordinating the inflammatory response, regulating the inflammatory gene expression [20]. In addition, phosphorylated STAT3 forms a dimer that can translocate into the nucleus and bind to the DNA, positively regulating the transcription of suppressor of cytokine signalling 3 (SOCS3) [18]. Interestingly, a positive regulation of Nfr2 activity has also been observed following the activation of α7 nAChR [3,21] (Figure 2A). This factor coordinates the cellular resistance to oxidants, activating the expression of antioxidant genes such as heme-oxigenase-1 (HO-1) [22]. The Nrf2 pathway has received increasing recognition as a key regulator of α7 nAChR-mediated neuroprotection. In fact, treatment with α7 nAChR agonist PNU-282987 considerably improves neuroprotective effects in mouse stroke model, suggesting an active involvement of the Nrf2 pathway in neuroprotection [21]. The pathway of Nfr2 is mediated by the activation of the PI3K/Akt signalling. α7 nAChR can activate the nuclear translocation of Nrf2, which can bind to DNA and regulate the expression of responsive antioxidant genes [3] (Figure 2A). These data demonstrate that through these different signal transduction pathways, α7 nAChR prevents the production of pro-inflammatory cytokines TNFα and IL-1β.

Similarly to that observed in the immune cells, in the CNS, it has been observed that the activation of α7 nAChR mediates an anti-inflammatory response only through the metabotropic pathway in mouse glial cells (i.e., microglia and astrocytes) [3,23,24]. Microglia-mediated inflammation is essential to the primary acute CNS immune response; however, this acute response must be resolved to prevent chronic activation. The activation of α7 nAChR in mouse microglia involves the activation of PLC via Gαq, inducing a release of Ca^2+^ from the intracellular stores (i.e., the endoplasmic reticulum) (Figure 2B) [25]. The increase in intracellular calcium mediates the decrease in phosphorylation, and, consequently, the activation of MAP kinases involved in neuroinflammation such as JNK, p38, and p44/42 [26,27]. These kinases regulate the synthesis and release of toxic and inflammatory molecules such as TNFα, IL-6, and nitric oxide (NO) in microglia [28,29] (Figure 2B).

In conclusion, these findings confirm that α7 nAChR stimulation mediates anti-inflammatory and antioxidant effects both in the immune system and in the brain.

## 4. α7 nAChR Effects in the Glial Cells

In rodents, nAChR subunits are detected during development, starting from embryonic stage 18 (E18), with an increase in expression during the post-natal early synaptogenesis events (P7–P14), having a central role in neuroblast migration and synapses formation [30], and then in neuronal growth and differentiation [31]. Evidence of the CNS neuroprotective role of nAChRs was obtained in the rat cerebral cortex, where NMDA receptors are accepted as predominantly responsible for glutamate cytotoxicity. Here, nicotine prevents glutamate neurotoxicity, and the neuroprotective effects are antagonized by mecamylamine, which is one of the main nAChR antagonists, but not by scopolamine, a muscarinic acetylcholine receptor antagonist, demonstrating that the neuroprotection is selectively mediated by nAChR activation [32]. Interestingly, nicotine significantly reverses glutamate cytotoxicity, whereas muscarine exacerbates it. Carbachol, acting on both nicotinic and muscarinic receptors, reduces glutamate cytotoxicity, although its efficacy is less evident than nicotine. These observations indicate how nAChRs and mAChRs play opposite effects on glutamate cytotoxicity [10,33]. Neuroprotective effects mediated by nAChRs, similar to the cerebral cortex, have been detected in other different areas of the brain, including the hippocampus [34], the striatum [35], the substantia nigra [36], and the spinal cord [37].

Although α7 nAChR is preferentially expressed in the presynaptic region, several pieces of evidence have demonstrated possible postsynaptic nicotinic signalling [3], and despite the well-characterized expression of nAChR in neuronal populations, their expression and relevance in glial cells have been only recently analysed. In the CNS, its expression has indeed been found in rat astrocytes [38] and mice microglia [23], where the “brain cholinergic anti-inflammatory pathway” produces neuroprotection, decreasing inflammatory reactions [3,39]. Moreover, astrocytes and microglia express acetylcholine acetyltransferase (ChAT) and several cholinergic receptors, giving these cells the ability to synthesise ACh and respond to cholinergic stimuli in an autocrine and/or paracrine manner [40].

Shytle and colleagues demonstrated that cultures of mice microglia cells express α7 nAChR transcript and protein [23]; α7 nAChR activation via nicotine or ACh exposure inhibits the production of TNFα with an increase in negative regulators of pro-inflammatory agents such as COX-2 and PGE2 in rat microglial cells [25,41]. Interestingly, nicotine’s effect on TNFα production and PGE2 release is counteracted by the specific α7 nAChR antagonist, α-bungarotoxin. Moreover, treatment with the α7 nAChR agonist PNU-282987 improves the response to neuroinflammation mediated by microglia in a mouse stroke model, leading to decreased infarct size and improved motor skills [21]. All these studies confirm that the anti-inflammatory processes are mediated by α7 nAChRs.

In addition to their role in microglia, α7 nAChRs are also expressed in astrocytes. Nicotine significantly decreases the release of pro-inflammatory cytokines such as IL-6, TNFα, and IL-1β from cultured human fetal astrocytes activated by recombinant IL-1β [42].

Moreover, the partial agonist GTS21 treatment significantly reduces the LPS-mediated secretion of inflammatory cytokines, its effect blocked by the α7 nAChR antagonist methyllycaconitine (MLA) or by α7 nAChR knockdown. Treatment with GTS21 also upregulates canonical Nrf2 antioxidant gene and protein, suggesting antioxidant properties of α7 nAChR also in mice astrocytes [24]. Nicotine carries out a protective effect on H_2_O_2_-induced astrocyte apoptosis and glial cell-derived neurotrophic factor (GDNF) regulation; this effect is counteracted by α7 nAChR selective antagonism in 1-methyl-4-phenyl-1,2,3,6-tetrahydropyridine (MPTP) mouse models [43]. The systemic administration of nicotine ameliorates acute MPTP-induced behavioral symptoms, preventing dopaminergic neurons’ degeneration through the inhibition of astrocytes and microglia activation in the substantia nigra pars compacta (SNpc) and blocking the decrease of GDNF in the striatum [43]. Similarly, in Alzheimer’s disease, the selective activation of α7 nAChRs expressed on hippocampal astrocytes counteracts the inflammation induced by amyloid β protein_1–42_ [44]. The expression and function of α7 nAChR in glial cells therefore clearly support its ability to significantly modulate neuroinflammation.

## 5. Role of α7 nAChR in the Control of Neuroinflammation in the CNS Pathologies

Neurodegenerative diseases are characterized by decreased cognitive functions and, similarly to aging, are a process causing a significant and progressive decline of the brain’s functions, affecting cognitive performance. The common characteristic shared by neurodegenerative pathologies and aging is the highly inflammatory state of the brain [45]. Histological post-mortem analysis of brain sections has, in fact, revealed the presence of activated microglia [46]. As previously reported, microglia and astrocytes’ inflammatory profiles are regulated by cholinergic receptors [47,48]. During brain development and aging, the distribution of cholinergic receptors changes significantly and a reduction of nicotinic receptors could significantly contribute to the cognitive decline and impairment of glial neuroprotective functions [49].

The risk factors of neurodegenerative diseases include aging, altered gene expression, oxidative stress, and systemic inflammation [50,51].

In Alzheimer’s disease (AD), the cognitive deficits are associated with alterations of the cholinergic system in brain regions involved in memory and learning functions, highlighting that cholinergic dysfunctions, mainly dependent on the α7 nAChRs decrease, are responsible for the dementia symptoms [52,53,54,55,56]. Interestingly, AD is characterized by β-amyloid (Aβ) neurotoxicity. However, in vivo, not only Aβ but also its modified forms can drive AD pathogenesis. One of these forms, iso-Aβ (containing an isomerized Asp7 residue), shows an increased neurotoxicity in vitro and stimulates amyloidogenesis in vivo [57].

Several data suggest the ability of Aβ to bind to α7 nAChRs. Consequently, the Aβ/α7 internalization causes the intracellular accumulation of Aβ, increasing neurotoxicity [53].

The neuroinflammation induced by Aβ fragments may be counteracted by the activity of α7 nAChRs expressed by the glial cells. In vitro studies have demonstrated that α7 nAChR activation with the selective agonist DMXBA promotes Aβ phagocytosis by cultured microglia cells [58]. On the other hand, AD-mice model in vivo studies have also demonstrated that α7 nAChR stimulation improves cognitive functions [59]. For instance, the use of α7 nAChR agonists that may displace the binding of Aβ to α7 nAChRs, reducing the Aβ internalization and neurotoxicity, could be considered relevant.

Parkinson’s disease (PD) is another frequent neurodegenerative disease, characterized by the loss of dopaminergic neurons in the mesencephalic area. Although the reduction in dopaminergic neurons is the first event characterizing this neuropathology, the subsequent degeneration of cholinergic system neurons in the basal forebrain contributes to dementia associated with PD [60]. In several studies based on the use of in vitro and in vivo mice models of PD, the use of 1-methyl-4-phenyl-1,2,3,6-tetrahydropyridine (MPTP) or LPS causes an increased number and activation of microglia and astrocytes, causing a strong neuroinflammation. Interestingly the systemic administration of nicotine in PD mice models improves the cognitive and motor abilities, reducing the loss of dopaminergic neurons, astrocyte activation in the brain and TNFα production and ERK1/3 activation [61]. Both effects are reverted by the α7 nAChR antagonist MLA. Conversely, the use of α7 nAChR agonist PNU-282987 protects microglia from apoptosis, increasing the anti-apoptotic protein Bcl-2, and decreasing caspase-3 activation [62].

The cholinergic activity also has a relevant role in the modulation of neuroinflammation in demyelinating disease [63].

Multiple sclerosis (MS) is an autoimmune pathology characterized by demyelination and chronic neuroinflammation. Several studies in EAE mice, one of the most frequently used MS mice models, have suggested how the treatment with cholinesterase inhibitors may reduce neuroinflammation and improve the motor and cognitive impairment. These effects are significantly counteracted by α7 nAChR antagonists [64,65,66]. In fact, α7 nAChRs play a relevant role in the immune system, where they control the number of dendritic cells and the proliferation of the autoreactive T cells [63]. Recently it has also been demonstrated that a cholinergic system dysfunction is also present in the immune system and in the brain of MS patients, suggesting that the altered immune response and neuroinflammation characterizing MS may be correlated to cholinergic system alterations [48,67,68,69,70].

Considering the role of α7 nAChR in the modulation of the cholinergic anti-inflammatory pathway [47], the expression of α7 subunits and the ability of nicotine in the modulation of the inflammatory cytokines were also evaluated in MS patients [71]. The nicotine stimulation of peripheral blood mononuclear cells (PBMC) derived from MS patients has demonstrated a reduced IL-1β and IL-17 production [71]. These data suggest how the non-neuronal components of a cholinergic system work in a paracrine or autocrine way, both in the brain and in the immune system, and also contribute to the modulation of inflammatory cytokines in MS patients.

Pharmacological studies also support a potential link between α7 nAChR and the pathophysiology of schizophrenia (SZ). By ligand-binding and immunohistochemical analyses, a decreased α7 nAChR expression in the hippocampus, cortex, and thalamus of schizophrenic patients has been demonstrated [72,73].

Although single nucleotide polymorphisms (SNPs) of the *CHRNA7* gene have not been found in SZ patients, multiple SNPs are found in the promoter region of the *CHRNA7* gene, which could affect the expression of the gene [74]. A moderate risk of developing schizophrenia may be also associated with the presence of 2bp deletion in exon 6 of *CHRFAM7A,* a duplicated form of *CHRNA7,* generating a premature stop coding sequencing that produces a shortened peptide. This dupΔα7 produces a dominant negative form that could interfere with the correct oligomerization process of the pentameric α7 nAChR and receptor functionality [75,76].

Nicotine administration enhances the sensorial deficit in schizophrenia, suggesting that the use of more selective ligands may have a clinical relevance in the treatment of the neurological dysfunction typically associated to this pathology [77].

## 6. α7 nAChR Neuropharmacology

The main features of α7 nAChRs include high Ca^2+^ permeability, a relatively low sensitivity to ACh, a high-affinity for α-BTX, and a relatively low affinity for nicotine.

Several selective ligands were initially developed and tested for their functionality on α7 nAChR, and their therapeutic potentiality was tested on mechanisms implicated in inflammation, memory, and behavioural disorders. Different drugs targeting the nAChRs are currently in the clinical trial stage on humans, and different α7 full agonists have been characterized [78]. In general, SEN 12333, PNU-282907, AR-R1777, and TC5619 bind the orthosteric site of the receptors, similarly to ACh (Table 1) [54].

Several racemic mixtures of spirocyclic derivatives of quinuclidinyl-Δ^2^-isoxazoline have been synthesized. The obtained compounds were then tested for their binding affinity for the neuronal α7 nAChRs (homomeric) and α4β2 (heteromeric), both in rats and humans. Among all, the racemic pair (±) -3-methoxy-1-bone-2,7-diaza-7,10-ethanospiro [4.5] dec-2-ene sesquifumarate is characterized by high affinity and selectivity levels for α7 nAChR in both binding and functional assays [79]. The (R)-(-)-enantiomer was then found to be the enantiomer with more pronounced biological activity, with a Ki value of 4.6 nM for rat and human α7 nAChRs [79]. This compound, called ICH3 [(R)-(‒)-3-methoxy-1-oxa-2,7-diaza-7,10-ethanospiro [4.5] dec-2-ene sesquifumarate], has the ability of selectively binding with the α7 receptors. This ability was confirmed by the use of α7 antagonist α-BTX in different rodent cell types [80,81].

Other studies have been also focused on nAChR partial agonists, ligands able to activate the ion channel with lower efficacy than the endogenous agonists (i.e., nicotine, GST-21) [82]. Among these ligands, S24795 (2-[2-(4-Bromophenyl)-2-oxoethyl]-1-methylpyridinium iodide) has been studied for AD [83]. Special attention has been focused on a new class of drugs called silent agonists [84], which produce very little channel activation but strong desensitizing (i.e., NS6740) [85].

nAChR activation can also occur via an allosteric site. The allosteric compounds can act as: (a) positive allosteric modulators (PAMs), able to potentiate currents only in the presence of the agonist; (b) allosteric agonists that activate the receptors in non-orthosteric sites; (c) negative allosteric modulators (NAMs), which act as channel blockers by binding to the orthosteric or allosteric site; and (d) silent allosteric modulators (SAMs), which have no effect on orthosteric agonist responses but block allosteric modulation [86,87].

The pharmacology of α7 nAChR is contributing to identify new potential therapeutic tools for the treatment of different nervous system pathologies. These drugs could be of great interest in counteracting neuroinflammation and helping the re-establishment of the nervous system homeostasis.

## 7. Conclusions

Neuroinflammation is a strategic process required to restore the homeostasis of the nervous system. Although this process is necessary to contrast infection, trauma, or damage produced by neurodegenerative or demyelinating diseases, prolonged inflammation can be detrimental for the neurons.

Acetylcholine is involved in the modulation of the central and peripheral inflammation since the immune system cells, as well as microglia and astrocytes, express cholinergic receptors.

In the last few decades, particular attention has been given to α7 nAChR considering its ability in the modulation of anti-inflammatory processes, reducing the expression of pro-inflammatory effectors and cytokines.

Significant drug discovery efforts have been devoted to α7 nAChR, and several promising ligands with high selectivity and minimal or no side effects have been developed in order to avoid receptor desensitization. Some of these molecules have shown a therapeutic relevance for the treatment of different neurodegenerative pathologies such as Alzheimer’s and Parkinson’s. Considering the effects of α7 nAChR activation in astrocytes and microglia in negatively modulating inflammatory cytokines and oxidant agents, the clinical therapeutic potential that the full and partial α7 nAChR agonists may play in the modulation of the neuroinflammation is clearly relevant. However, considering the large distributions of these receptors inside and outside the CNS, the use of these pharmacological ligands could present some limitations, which should not be underestimated. The research for new α7 nAChR selective agonists is still on-going, trying to reduce or minimize the side effects associated.

## Figures and Tables

**Figure 1 ijms-22-04912-f001:**
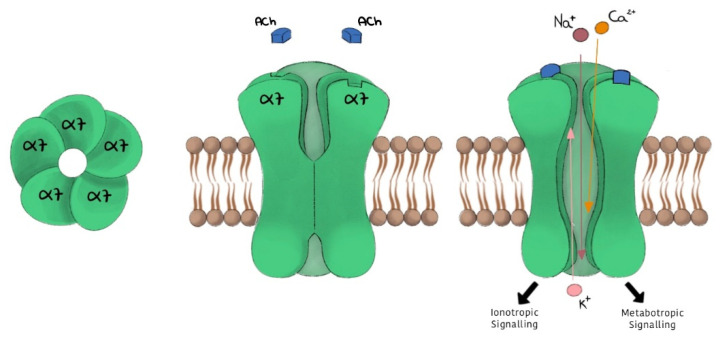
Schematic representation of α7 nAChR.

**Figure 2 ijms-22-04912-f002:**
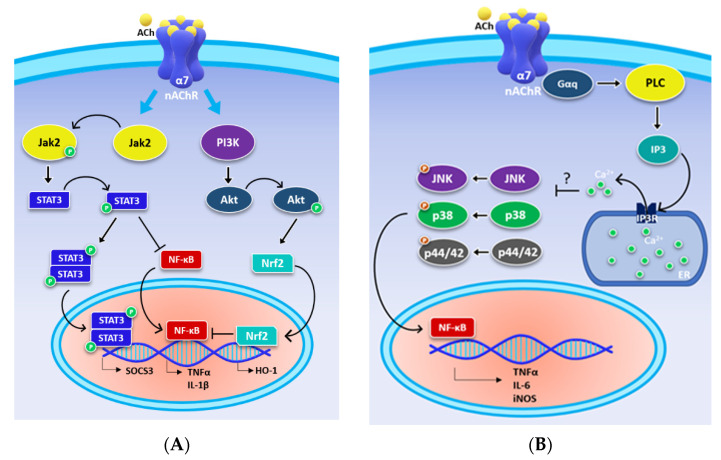
Metabotropic signalling pathway mediated by α7 nAChR activation. (**A**) The activation of Jak 2 and PIK3 initiates the signal transduction cascade, leading to the modulation of inflammatory processes characteristic of the cholinergic anti-inflammatory pathway in mouse macrophages. (**B**) In microglia, the α7 nAChR activation of PLC, mediated by Gαq, induces the release of Ca^2+^ from the endoplasmic reticulum (ER). IP3, produced by PLC, binds to the IP3 receptor (IP3R) present on the ER in mouse microglia. This pathway decreases the phosphorylation, and therefore the activation, of p38, p44/42, and c-jun N-terminal kinase (JNK) and MAP kinases involved in neuroinflammation.

**Table 1 ijms-22-04912-t001:** Full and partial α7 nAChR agonists.

Nicotinic Agonists	Receptor Selectivity	Ki
SEN 12333	Full agonist α7 subunit	260 nM
PNU-282907	Full agonist α7 subunit	27 nM
PNU-120596	Full agonist α7 subunit	0.9 µM
TC 5619	Full agonist α7 subunit	1 nM
ICH3	Partial agonist α7 subunit	4.6 nM
S24795	Partial agonist α7 subunit	34 nM
A-582941	Partial agonist α7 subunit	16 nM

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
