# Peer review of "Cholinergic Modulation of Neuroinflammation: Focus on α7 Nicotinic Receptor"

_ijms, 2021, doi:10.3390/ijms22094912_

Round 1
Reviewer 1 Report
Major concerns
- The manuscript has typographical errors and a number of inconsistencies that must be addressed to ensure that it is fluently written and can convey the information clearly to the readers. I strongly suggest that the manuscript be extensively edited and peer reviewed, preferably by researchers outside the research area of the authors of the manuscript. Another issue is that many small paragraphs are presented as facts, and the logical flow is poor from one to the next.
- More care should be taken when citing literature references to ensure that the original manuscript is cited to give credit to the contribution itself, rather than to a review citing the original contribution. For example, line 58 is credited to reference 1; however, this seminal contribution is well recognized as being published by the group in reference 2.
- The content of section 3 does not represent the title of the section. The title suggests that the distribution of the α7-nAChR will be presented at the CNS level; however, this is not the case. In fact, section 3 could be reformatted into an elegant table that would show the areas of the CNS that express α7-nAChR in humans, mice, and rats (if the intention is to compare these models against humans). Finally, this section ignores the fact that α7-nAChR has also been identified in interneurons.
- Throughout the manuscript, there is no indication whether the data presented comes from humans, rats, mice, or any other model. This information should be clarified throughout the manuscript.
- The information presented in Section 4 is terribly disorganized, so it is very difficult to follow the information throughout the section. For example, no indication is provided regarding whether the information comes from studies with humans or animal cells. Another example is the inclusion of information about peripheral macrophages, instead of focusing on microglia, since the manuscript’s focus is on the CNS.
- Another worrying point is that Figure 2 does not recapitulate the signaling molecules/pathways that are mentioned in section 4, where the figure is cited. Nor is it indicated whether the signaling presented in Figure 2 represents the activity of α7-nAChR acting as an ionotropic receptor or as a metabotropic receptor. Undoubtedly, this review, and the target readers in this field, would benefit substantially from the inclusion of a two-panel summary figure showing the signaling molecules/pathways that have been identified so far for the α7-nAChR acting as an ionotropic or metabotropic receptor in the CNS. This would add greatly to the impact of the paper.
- The penultimate paragraph (line 274), presents information about α7-nAChR and schizophrenia. This small paragraph would benefit significantly by including the existing information on the partial duplication of α7-nAChR that exists in humans and that has been shown to be related to schizophrenia (Kalmady et al., 2018, Sinkus et al., 2009, Severance et al., 2009, Iwata et al., 2007, DeLuca et al., 2006, and Flomen et al., 2006).
- Section 7 is very fragmented and again does not flow well. In fact, the heading should read neuropharmacology instead of "pharmacology." More importantly, no information is provided to indicate whether the mentioned drugs have been tested in cells derived from the CNS or in animal models focused on the CNS.
- The conclusion section is a perfect example of the fragmentation of the information presented in this manuscript. It includes many short paragraphs that seem more like facts than part of a cohesive conclusion.
Minor concerns
- Figure 1 has room for improvement. I strongly recommend that the hydrophobic tails be added to the phospholipids in the membrane. The ions that permeate through the channel could also be presented to add functionality and dynamism to the figure.
- The nomenclature presented in the manuscript for the receptor must be standardized. Is it α7 receptor or α7 nAChR?
- Table 1 does not indicate which are partial and which are full agonists of α7-nAChR. This information is essential and must be indicated.
Author Response
The authors are grateful for the comments and suggestions
Rev 1
Major concerns
- The manuscript has typographical errors and a number of inconsistencies that must be addressed to ensure that it is fluently written and can convey the information clearly to the readers. I strongly suggest that the manuscript be extensively edited and peer reviewed, preferably by researchers outside the research area of the authors of the manuscript. Another issue is that many small paragraphs are presented as facts, and the logical flow is poor from one to the next.
We have largely revised the text. We have moved sentences and paragraphs in order to improve the reading and understanding of the various contents
- More care should be taken when citing literature references to ensure that the original manuscript is cited to give credit to the contribution itself, rather than to a review citing the original contribution. For example, line 58 is credited to reference 1; however, this seminal contribution is well recognized as being published by the group in reference 2.
The reference has been corrected and new references have been added
- The content of section 3 does not represent the title of the section. The title suggests that the distribution of the α7-nAChR will be presented at the CNS level; however, this is not the case. In fact, section 3 could be reformatted into an elegant table that would show the areas of the CNS that express α7-nAChR in humans, mice, and rats (if the intention is to compare these models against humans). Finally, this section ignores the fact that α7-nAChR has also been identified in interneurons.
The section 3, now is the section 4. The intention of the authors is not to compare the distribution of α7-nAChR in different area of CNS. Our intent is to highlight how the expression of α7-nAChR is not exclusive of the neurons, but that it is present also in the glial cells. The title of the paragraph has been chamged
- Throughout the manuscript, there is no indication whether the data presented comes from humans, rats, mice, or any other model. This information should be clarified throughout the manuscript.
We have specified the animal model in the various paragraphs
- The information presented in Section 4 is terribly disorganized, so it is very difficult to follow the information throughout the section. For example, no indication is provided regarding whether the information comes from studies with humans or animal cells. Another example is the inclusion of information about peripheral macrophages, instead of focusing on microglia, since the manuscript’s focus is on the CNS.
Section 4 has been completely rewritten. The focus of this paragraph is to describe the metabotropic signallings active in glial cells. We have described the metabotropic pathways downstream alpha-7 activation in macrophages to compare them with the pathways activated in microglia. We have specified the animal model on which the studies were conducted
- Another worrying point is that Figure 2 does not recapitulate the signaling molecules/pathways that are mentioned in section 4, where the figure is cited. Nor is it indicated whether the signaling presented in Figure 2 represents the activity of α7-nAChR acting as an ionotropic receptor or as a metabotropic receptor. Undoubtedly, this review, and the target readers in this field, would benefit substantially from the inclusion of a two-panel summary figure showing the signaling molecules/pathways that have been identified so far for the α7-nAChR acting as an ionotropic or metabotropic receptor in the CNS. This would add greatly to the impact of the paper.
The figure 2 has been modified. We have generated two panels in which describes the metabotropic pathways describe in the section 4. We have emphasised the metabotropic signals, because prevalent in non neuronal cells (microglia and astrocytes)
- The penultimate paragraph (line 274), presents information about α7-nAChR and schizophrenia. This small paragraph would benefit significantly by including the existing information on the partial duplication of α7-nAChR that exists in humans and that has been shown to be related to schizophrenia (Kalmady et al., 2018, Sinkus et al., 2009, Severance et al., 2009, Iwata et al., 2007, DeLuca et al., 2006, and Flomen et al., 2006).
We have enclosed a short description of the duplicated form of alpha.7 and its implication in schizophrenia
- Section 7 is very fragmented and again does not flow well. In fact, the heading should read neuropharmacology instead of "pharmacology." More importantly, no information is provided to indicate whether the mentioned drugs have been tested in cells derived from the CNS or in animal models focused on the CNS.
This paragraph has been modified. The title of the paragraph has been changed and the animal models have been cited
- The conclusion section is a perfect example of the fragmentation of the information presented in this manuscript. It includes many short paragraphs that seem more like facts than part of a cohesive conclusion.
The Conclusion have been modified
Minor concerns
- Figure 1 has room for improvement. I strongly recommend that the hydrophobic tails be added to the phospholipids in the membrane. The ions that permeate through the channel could also be presented to add functionality and dynamism to the figure.
The figure 1 has been modified according to the suggestions
- The nomenclature presented in the manuscript for the receptor must be standardized. Is it α7 receptor or α7 nAChR?
We have uniformed the nomenclature of the receptor
- Table 1 does not indicate which are partial and which are full agonists of α7-nAChR. This information is essential and must be indicated.
The table 1 has been completed
Reviewer 2 Report
Please, correct in line 147 [34]. In this way it prevents the production of pro-inflammatory cytokines such us TNFα
Extra dots in headings in lines 65, 165, 281
Paragraph 224-239 lacks the following relevant and important information:
In vivo not only Aβ but also its modified forms can drive AD pathogenesis. One of these forms, iso-Aβ (containing an isomerized Asp7 residue), shows an increased neurotoxicity in vitro and stimulates amyloidogenesis in vivo. [Barykin, E.P., Garifulina, A.I., Kruykova, E.V., Spirova, E.N., Anashkina, A.A., Adzhubei, A.A., Shelukhina, I.V., Kasheverov, I.E., Mitkevich, V.A., Kozin, S.A. and Hollmann, M., 2019. Isomerization of Asp7 in beta-amyloid enhances inhibition of the α7 nicotinic receptor and promotes neurotoxicity. Cells, 8(8), p.771.]
Author Response
The authors are grateful for the comments and suggestions
Rev 2
Please, correct in line 147 [34]. In this way it prevents the production of pro-inflammatory cytokines such us TNFα
Thank you, we have corrected
Extra dots in headings in lines 65, 165, 281
Corrected
Paragraph 224-239 lacks the following relevant and important information:
In vivo not only Aβ but also its modified forms can drive AD pathogenesis. One of these forms, iso-Aβ (containing an isomerized Asp7 residue), shows an increased neurotoxicity in vitro and stimulates amyloidogenesis in vivo. [Barykin, E.P., Garifulina, A.I., Kruykova, E.V., Spirova, E.N., Anashkina, A.A., Adzhubei, A.A., Shelukhina, I.V., Kasheverov, I.E., Mitkevich, V.A., Kozin, S.A. and Hollmann, M., 2019. Isomerization of Asp7 in beta-amyloid enhances inhibition of the α7 nicotinic receptor and promotes neurotoxicity. Cells, 8(8), p.771.]
This sentence and the reference suggested have been enclosed in the manuscript
Round 2
Reviewer 1 Report
- Check the word signalling, it should read "signaling". It must be corrected throughout the manuscript, including in Figure 1.
- To maintain consistency, it must be decided whether the authors are going to use Ca2+ or Ca++. Arrange throughout the manuscript and in Figure 1.
- On lines 121, 348, 62, 65, 103, and 278, α7 nAChR should be placed instead of α7 receptor.
- On line 157, please put a space between α7 and nAChR.
- The two-line paragraph (277 and 278) should be reconciled with the paragraph above or below. The same for the paragraph contained in lines 287, 288, and 289, the latter should be included in some of the major paragraphs.
- Check line 52. I don't think the concept was defined properly. It must be α7 nAChR instead of nAChRs (also it is plural, it should be singular).
- Line 183 should read Figure 2, not Figure 1.
- Line 337 should read CHRNA7, not CHRA7.
- Line 408 should read α7 nAChR, not "α-7 nicotinic".
- It is strongly recommended that Table 1 include a column that gives credit to the manuscript that provides the Ki values.
Author Response
Reply to reviewer 1 –R1
Thank you for your suggestions. We have modified the manuscript and modified the figures according to the present comments
- Check the word signalling, it should read "signaling". It must be corrected throughout the manuscript, including in Figure 1.
Corrected
- To maintain consistency, it must be decided whether the authors are going to use Ca2+ or Ca++. Arrange throughout the manuscript and in Figure 1
OK . we have changed Ca 2+ in the figure 1 and 2B
- On lines 121, 348, 62, 65, 103, and 278, α7 nAChR should be placed instead of α7 receptor. Changed
- On line 157, please put a space between α7 and nAChR.
Changed.
- The two-line paragraph (277 and 278) should be reconciled with the paragraph above or below. The same for the paragraph contained in lines 287, 288, and 289, the latter should be included in some of the major paragraphs.
The sentence has been modified
- Check line 52. I don't think the concept was defined properly. It must be α7 nAChR instead of nAChRs (also it is plural, it should be singular).
changed.
- Line 183 should read Figure 2, not Figure 1.
Corrected
- Line 337 should read CHRNA7, not CHRA7.
Corrected
- Line 408 should read α7 nAChR, not "α-7 nicotinic".
Corrected
- It is strongly recommended that Table 1 include a column that gives credit to the manuscript that provides the Ki values.
The Ki values were already present in Table 1